

# Variation in floral morphology, histochemistry, and floral visitors of three sympatric morning glory species

Awapa Jirabanjongjit[1,*], Alyssa B. Stewart[1,*], Natthaphong Chitchak[1], Chakkrapong Rattamanee[2] and Paweena Traiperm[1]

[1] Department of Plant Science, Faculty of Science, Mahidol University, Bangkok, Thailand
[2] 37/1 Moo4 Saocha-Ngok, Chachoengsao, Thailand
[*] These authors contributed equally to this work.

## ABSTRACT

Three morning glory species in the genus *Argyreia* Lour., *A. lycioides* (Choisy) Traiperm & Rattanakrajang, *A. mekongensis* Gagnep & Courchet, and *A. versicolor* (Kerr) Staples & Traiperm, were found co-occurring and co-flowering. *Argyreia mekongensis* and *A. versicolor* are rare, while *A. lycioides* is near threatened and distributed throughout Myanmar and Thailand. We investigated key floral characters (floral morphology and phenology, as well as the micromorphology of the floral nectary disc and staminal trichomes) and screened for important chemical compounds hypothesized to contribute to pollinator attraction. Our findings demonstrate that some aspects of floral morphology (*e.g.*, corolla size, limb presence, and floral color) of the three studied congeners exhibit significant differences. Moreover, pollinator composition appears to be influenced by floral shape and size; morning glory species with wider corolla tubes were pollinated by larger bees. The morphology of the floral nectary disc was similar in all species, while variation in staminal trichomes was observed across species. Glandular trichomes were found in all three species, while non-glandular trichomes were found only in *A. versicolor*. Histochemical results revealed different compounds in the floral nectary and staminal trichomes of each species, which may contribute to both floral attraction and defense. These findings demonstrate some segregation of floral visitors among sympatric co-flowering morning glory species, which appears to be influenced by the macro- and micromorphology of flowers and their chemical compounds. Moreover, understanding the floral morphology and chemical attractants of these sympatric co-flowering *Argyreia* species may help to maintain their common pollinators in order to conserve these rare and endangered species, especially *A. versicolor*.

Corresponding author
Paweena Traiperm,
paweena.tra@mahidol.edu

# INTRODUCTION

Much of floral evolution is driven by pollinator attraction (*Fenster et al., 2004*) and pollination efficiency (*Stewart et al., 2022*). To attract pollinators, floral morphology and phenology are prominent and important features (*Bobisud & Neuhaus, 1975*; *Schemske, 1981*), including flower size, color, and scent (*Rathcke, 1983*; *Waser & Price, 1983*; *Spaethe,*

*Tautz & Chittka, 2001*; *Waser & Ollerton, 2006*; *Willmer, 2011*; *Hassa, Traiperm & Stewart, 2020*; *Hassa, Traiperm & Stewart, 2023*). For example, large flowers tend to be favored and selected for by insects since they are more visible (*Chittka & Raine, 2006*; *Naug & Arathi, 2007*; *Benitez-Vieyra et al., 2010*) and reduce search time during pollinator foraging (*Spaethe, Tautz & Chittka, 2001*). Similarly, *Thompson (2001)* demonstrated the importance of floral size or display on variation in visitation patterns between different insect types; hawkmoths and butterflies both preferred larger floral displays. Floral color is another important trait for pollinator attraction since different pollinator species have different spectral receptor cells; for example, hummingbirds tend to select red flowers because of high chromatic contrast to the background (*Herrera et al., 2008*), whereas bees are uncommon visitors to red flowers given their lower sensitivity to red wavelengths (*Bergamo et al., 2016*). Flowering time and floral phenology also influence pollination and reproductive success (*Evans, Smith & Gendron, 1989*; *Elzinga et al., 2007*). Phenology has been shown to affect pollinator variation and effectiveness, which can impact fitness, as has been demonstrated through reduced seed mass throughout the flowering season (*Gallagher & Campbell, 2020*).

While floral morphology and phenology tend to be highly prominent and easily studied, there are other floral traits that are less visible, yet are also important in plant–pollinator interactions. For example, the floral nectary is an important floral organ as it provides the primary reward for many pollinators, nectar (*Pacini & Nicolson, 2007*; *Irwin et al., 2010*). The floral nectary is important not only in terms of providing food resources for pollinators (*Simpson & Neff, 1981*; *Proctor, Yeo & Lack, 1996*; *Neiland & Wilcock, 1998*; *Nicolson, 2007a*) but also for manipulating pollinator behavior (*Bailey et al., 2007*). However, nectar not only attracts pollinators but also nectar robbers (*Inouye, 1980*). Consequently, many plant species have evolved features that only allow visitation by specific pollinators (*e.g.*, pollination syndromes; *Fenster et al., 2004*). Nectar production is one trait that has an important role in confining the range of visitors to species that benefit the plants (*Irwin, Adler & Agrawal, 2004*). In angiosperms, nectar is dominated by sugars, although the proportions of different sugar types vary according to species (*Baker, 1982*; *Freeman, Worthington & Jackson, 1991*; *Stiles & Freeman, 1993*). Other compounds in nectar have also been investigated, such as terpenes and lipids (*Nicolson, 2007b*). Terpenes are present in the nectar of diverse plant species and are generally considered to be attractants (*Hammer & Menzel, 1995*; *Raguso, Light & Pickersky, 1996*; *Plepys et al., 2002*; *Cunningham et al., 2004*). Plants also produce lipids in many parts, including pollen and nectar, which provide nutrition and serve as rewards for pollinators, especially oil bee pollinators (*e.g.*, *Martins & Alves-dos Santos, 2013*; *Rabelo et al., 2014*). While floral morphology is an important component for predicting pollination syndromes, other characters should also be considered, such as flowering phenology, anthesis start time and duration, and chemical compounds in nectar that are certainly associated with pollination and pollinator activities (*Southwick, Loper & Sadwick, 1981*; *Baker & Baker, 1983*; *Pleasants, 1983*; *Waser et al., 1996*; *Galetto & Bernardello, 2005*; *Ollerton et al., 2009*; *Bobrowiec & Oliveira, 2012*). Additionally, macroevolutionary studies can reveal associations between nectar traits and pollinator types, for example, finding similar nectar properties in plants

that are visited by the same pollinators (*Faegri & Van der Pijl, 1979*; *Proctor, Yeo & Lack, 1996*). Floral chemistry can mediate interactions with pollinators, pathogens, and/or herbivores, and therefore influence plant fitness (*Strauss & Whittall, 2006*; *Irwin et al., 2010*; *Good et al., 2014*).

Another plant organ important in mediating plant-animal interactions is the trichome. Trichomes initially originate from expansions or appendages of the epidermis (*Evert, 2006*), and have diverse biological functions, such as in herbivore defense, pollinator attraction, or tissue protection and maintenance (*Nihoul, 1993*; *Van Dam & Hare, 1998*; *Kennedy, 2003*; *Moyano, Cocucci & Sersic, 2003*; *Simmons & Gurr, 2005*; *Liu et al., 2006*; *Horgan et al., 2007*; *Gonzales et al., 2008*; *Romero, Souza & Vasconcellos-Neto, 2008*; *Nonomura et al., 2009*; *Kang et al., 2010*; *Karabourniotis et al., 2020*), which may be due to the synthesis and storage of biologically active metabolites (*Alonso et al., 1992*; *Antonious, 2001*; *Iijima et al., 2004*; *Siebert, 2004*; *Deschamps et al., 2006*; *Nagel et al., 2008*; *Wang et al., 2008*; *Biswas et al., 2009*; *Sallaud et al., 2009*; *Luo et al., 2010*). Trichomes are typically found on vegetative and reproductive organs, such as leaves, stems, petals, petioles, peduncles, and seeds (*Wagner, Wang & Shepherd, 2004*). Trichomes can also be found on staminal filaments, although they have been less studied and their function in many cases is still unclear. Staminal trichomes have been reported in five species in the genus *Teucrium* L. (Lamiaceae) (*Bini Maleci & Servettaz, 1991*) and in some species of *Argyreia* Lour. (*Van Ooststroom, 1943*; *Van Ooststroom, 1945*; *Van Ooststroom, 1950*; *Van Ooststroom, 1952*; *Hoogland, 1952*; *Van Ooststroom & Hoogland, 1953*; *Chitchak et al., 2018*; *Chitchak, Stewart & Traiperm, 2024*) and *Rivea* Choisy (*Chitchak, Stewart & Traiperm, 2022*; *Chitchak, Stewart & Traiperm, 2024*). Staminal trichomes have been proposed to contribute to pollinator attraction (*Jirabanjongjit et al., 2021*; *Chitchak, Stewart & Traiperm, 2022*).

Floral traits such as corolla color, size, and shape, as well as the chemical composition of the nectary and staminal trichomes, can all influence floral visitors. In co-flowering communities, floral traits and flowering time are major factors that influence plant–pollinator interactions and the degree to which plants have to share or compete for pollinators (*McEwen & Vamosi, 2010*; *Suárez-Mariño et al., 2022*). Most research shows that competition for pollination among co-flowering species generally has negative effects on plant fitness, particularly for rare plant species (*Levin & Anderson, 1970*; *Johnson, Dutt & Levine, 2022*). Such competition for pollination often leads to the evolution of reproductive isolation mechanisms, such as sympatric species diverging in flowering phenology or morphology (*Levin, 1971*; *Liu & Huang, 2013*; *Ramírez-Aguirre et al., 2019*). However, some studies have shown that highly similar floral traits and/or overlapping flowering periods can enhance pollinator attraction and increase pollinator sharing, resulting in a generalized plant–pollinator network structure (*Schemske, 1981*; *Sargent & Ackerly, 2008*; *De Jager, Dreyer & Ellis, 2011*; *Lázaro et al., 2020*; *Suárez-Mariño et al., 2022*). Moreover, high floral similarity with low flowering overlap can allow plants to benefit from shared pollinator attraction without incurring the costs of interspecific pollen transfer (*Bizecki Robson, 2013*). Interspecific pollen transfer can also be reduced, even among species with overlapping flowering phenologies, through mechanisms such as high pollinator constancy

(*De Jager, Dreyer & Ellis, 2011*) or differential pollen placement (*Huang, Liu & Huang, 2015*; *Stewart & Dudash, 2016*).

In this study we examined three sympatric *Argyreia* species (*A. lycioides*, *A. mekongensis*, and *A. versicolor*) that have seemingly similar floral morphologies and phenologies, which raises the question about the extent to which these species share, partition, or compete for pollinators. Two of these species, *A. versicolor* and *A. mekongensis*, are rare—especially the former, which is extremely rare and near extinction (*Staples et al., 2021*; *Jirabanjongjit et al., 2024*)—while the status of *A. lycioides* is Near Threatened (*Rattanakrajang et al., 2022*). Some aspects of their reproductive biology are already known, *i.e.,* floral visitor composition and anthesis of *A. mekongensis* and *A. versicolor* (*Jirabanjongjit et al., 2024*), and the present study aims to (1) gather similar floral visitor and anthesis data for *A. lycioides*, (2) study the floral histochemistry of the three species, and (3) compare the three species in terms of floral morphology, phenology and histochemical characteristics in order to understand how they relate to pollinator attraction and reward. Assessing the macromorphological, micromorphological, histochemical, and phenological differences between these three *Argyreia* species will inform our understanding of how these sympatric congeners share, partition, or compete for pollinators, which is vital for their conservation.

## MATERIALS & METHODS

### Study site and study species

This study was conducted on the campus of Burapha University in Sa Kaeo province, Thailand, where the natural distributions of our study species intersect. The study area is characterized as a lowland watershed with undulating plains (Sa Kaeo Provincial Office, https://sakaeopao.go.th/location/, January 2022) that are primarily covered with deciduous dipterocarp forest (A. Jirabanjongjit, 2019, pers. obs.). The local climate is tropical with seasons influenced by two monsoons, resulting in three distinct seasons: summer, rainy, and winter (Thai Meteorological Department, http://www.climate.tmd.go.th, January 2022). The summer season begins in mid-February to mid-May, and the region experiences high temperatures ranging from 25 to 35 °C, coupled with elevated humidity levels and minimal precipitation. Summer is followed by the rainy season (mid-May until mid-October), during which temperatures decrease slightly and precipitation increases substantially, particularly in August and September, which is the start of the flowering season for our three study species. The winter season spans from mid-October until mid-February, during which temperatures are cooler but humidity remains relatively high (Thai Meteorological Department, http://www.climate.tmd.go.th, January 2022).

The populations for each of our three study species were extremely small. We found only two individuals of *A. versicolor* (located approximately 50 m apart), nine individuals of *A. mekongensis* (located approximately 500–1,000 m apart), and 20 individuals of *A. lycioides* (located around 5 m apart). Both *A. mekongensis* and *A. versicolor* are woody twiners (*Staples & Traiperm, 2010*), commonly found growing on wild plants, *A. mekongensis* around 1.5 m from the ground and *A. versicolor* around 5 m from the ground. In contrast, *A. lycioides* is a woody shrub typically 0.5–3 m tall (*Staples, 2010*;

*Rattanakrajang, Traiperm & Staples, 2018*; *Rattanakrajang et al., 2022*). We observed the floral morphology and flowering phenology of all three species during their flowering periods in 2019 and 2020.

*Argyreia versicolor* (Kerr) *Staples & Traiperm (2010)* has an ovate leaf shape with an axillary inflorescences comprising 7–12 flowers per inflorescence. The flowers are composed of large pinkish green bracts. The corolla is white, tubular-campanulate, ca. five cm long, with purple-dotted limbs (*Staples & Traiperm, 2010*). The flowers have two styles which are significantly longer than the stamens. Flowers are self-incompatible (*Jirabanjongjit et al., 2024*). The fruits are globose berries (*Jirabanjongjit et al., 2024*).

*Argyreia mekongensis* Gagnep & Courchet has an elliptic to broadly oblong leaf shape with axillary inflorescences comprising 5–7 flowers per inflorescence (*Staples & Traiperm, 2010*). The flowers are accompanied by large pale greenish bracts that remain even after fruits are mature (*Staples & Traiperm, 2010*). The corolla, which is around five cm in length, is pure white, campanulate, and ca. 5–6 cm long (*Staples & Traiperm, 2010*). The flowers contain five stamens and two stigma lobes. Flowers are self-incompatible (*Jirabanjongjit et al., 2024*). The fruits are globose berries (*Staples & Traiperm, 2010*).

*Argyreia lycioides* (Choisy) Traiperm & Rattanakrajang has an elliptic-lanceolate leaf shape with solitary flowers (*Rattanakrajang et al., 2022*). The flowers are composed of four small greenish bracts. The corollas are around $1-2.5$ cm long, urceolate in shape, and pale yellowish-white in color (*Rattanakrajang et al., 2022*). The flowers have five stamens and two stigma lobes. The mating system is unknown. The fruits are capsules with a persistent calyx (*Rattanakrajang et al., 2022*).

## Floral characters and flowering phenology

Data collection was conducted during the flowering period of all three study species over two years of observation (2019–2020). The floral characters of *A. lycioides* (40 flowers), *A. mekongensis* (53 flowers) and *A. versicolor* (66 flowers) were observed, measured and recorded following terminology from the Kew glossary (*Beentje, 2010*) and the Flora of Thailand (Convolvulaceae) (*Staples, 2010*), as well as from recent studies of the three species (*Staples & Traiperm, 2017*; *Rattanakrajang et al., 2022*). The flowering phenology of each study species was assessed from field work, herbarium specimens, and relevant literature (*Staples, 2010*; *Rattanakrajang, Traiperm & Staples, 2018*; *Rattanakrajang et al., 2022*).

We assessed 15 characters predicted to influence plant interactions with pollinators: habit, flower arrangement (inflorescence or solitary), corolla shape, corolla length, corolla tube diameter, corolla limb presence, corolla color, phenology, floral anthesis, floral longevity, stamen position (included or excluded), stamen length, pistil length, and the staminal trichome densities of glandular and non-glandular trichomes (Table 1). Staminal trichome morphology was described based on *Chitchak, Stewart & Traiperm (2024)* and the distribution and density of trichomes were assessed by examining the apex, middle, and bottom of trichome distribution along the stamens. One-way ANOVA (package "multcomp") was conducted in R version 4.1.2 (*R Core Team, 2022*) to assess whether the three study species differed in terms of four quantitative characters of interest: corolla

length, corolla tube diameter, stamen length, and pistil length. For significant characters, Tukey's tests were used for post hoc analyses (package "multcomp").

## Floral anthesis and visitor observation

The floral anthesis and floral visitors of *A. mekongensis* (observed from 25 flowers from five plants across four days in 2019 and 29 flowers from five plants across seven days in 2020) and *A. versicolor* (22 flowers from two plants across four days in 2019 and 44 flowers from the single study plant remaining across eight days in 2020) were recently reported (*Jirabanjongjit et al., 2024*), but no such records for *A. lycioides* were found.

To determine the floral visitors of *A. lycioides*, we used action cameras (Xiaomi YI Z15; Xiaomi, Beijing, China) placed in front of mature flowers to capture animal visits. During 2019, we recorded 16 flowers from four individuals across four days, while in 2020 we observed 24 flowers from four individuals across seven days. We did not collect floral visitors to avoid disturbing subsequent animal visits and to avoid damaging flowers with sweep nets. All footage was reviewed and floral visitors were identified to the lowest taxonomic level possible with help from a local entomologist (see Acknowledgments).

Animal visitors were categorized based on their behavior at flowers as follows: potential pollinators (those that contacted both stigmas and anthers), florivores (those that consumed any parts of the flowers), and visitors/nectar robbers (those that visited flowers without contacting reproductive structures). The visitation rates of animal visitors were compared using R version 4.1.2 (*R Core Team, 2022*). Linear mixed modelling was performed using package 'lme4' where visitation rate was the response variable, animal taxon was a fixed factor, and plant individual was a random factor. Models were assessed using nested likelihood ratio tests (package 'stats'). Turkey's post hoc test (package 'emmeans') was used for comparing factor levels.

Permission to work with animals was granted by MUSC-IACUC (Faculty of Science, Mahidol University-Institutional Animal Care and Use Committee) (Protocol numbers MUSC60-037-387 and MUSC63-031-539).

## Histochemical examination

Histochemical techniques were used to detect the presence of chemical compounds of interest in the floral nectary discs (found surrounding the ovary at the base of the corolla) and staminal trichomes (found around the bases of the filaments). We examined 10 flowers per study species. Nectary discs were free hand-sectioned both transversally and longitudinally. Staminal trichomes were removed from the filament base. All sample specimens were treated with the following histochemical assays: NADI reagent to test for terpenes (*David & Carde, 1964*; *Olaranont et al., 2018*), Sudan Black B and Sudan III to test for lipids (*Brundrett, Kendrick & Peterson, 1991*), and Naturstoff to test for flavonoids (*Olaranont et al., 2018*; *Tattini et al., 2000*). Samples stained with NADI reagent, Sudan Black B, and Sudan III were examined under a light microscope (Olympus CX21 equipped with a Sony 6400 digital camera, Tokyo, Japan) and samples stained with Naturstoff were examined under a fluorescence microscope (Olympus BX53 with a DP73 camera set, Waltham, MA, USA) with a 436-nm exciter filter.

Jirabanjongjit et al. (2024), *PeerJ*, DOI 10.7717/peerj.17866

**Table 1** Floral characters, habit, phenology and trichome characters.

| Characters | Species | | |
| --- | --- | --- | --- |
| | *A. versicolor* | *A. mekongensis* | *A. lycioides* |
| Habit | Liana | Liana | Shrub |
| Flower arrangement | Inflorescence | Inflorescence | Solitary flower |
| Corolla shape | Campanulate | Campanulate | Campanulate |
| Corolla length (mm) | 56.6 ± 1.9 | 49.5 ± 1.7 | 30.1 ± 0.4 |
| Corolla tube diameter (mm) | 28.1 ± 1.0 | 23.8 ± 1.2 | 20.5 ± 0.4 |
| Corolla limb presence | Present | Present | Absent or very small |
| Corolla color | White corolla with purple limb | White with small brownish dots | Greenish white with dark purple dots inside |
| Flowering period (Phenology) | August to December | Late August to early December | Early September to late October |
| Floral anthesis | 05.30 (complete around 07.00) [*] | 05.00 (complete around 07.00–08.00)[*] | 05.00 (complete around 08.00–09.00) |
| Floral longevity | 15–16 h [*] | 36–40 h [*] | 36–40 h |
| Stamen projection relative to corolla limb | Included | Included | Included |
| Stamen length (mm) | 22.5 ± 0.3 | 17.8 ± 0.8 | 15.8 ± 0.4 |
| Glandular staminal trichome density (per mm$^2$) | 36.1 ± 2.3 | 32.7 ± 5.6 | 27.0 ± 4.0 |
| Non-glandular staminal trichome density (per mm$^2$) | 5.1 ± 0.5 | N/A | N/A |
| Pistil length (mm) | 39.7 ± 1.0 | 28.3 ± 0.9 | 17.3 ± 0.3 |

**Notes.**

All numerical data are presented as mean ± standard error (SE).

*Data obtained from *Jirabanjongjit et al. (2024)*.

## RESULTS

### Floral characters and flowering phenology

Both *A. versicolor* (Figs. 1A, 1B) and *A. mekongensis* (Figs. 1C, 1D) are lianas that produce several flowers per inflorescence, with flowers exhibiting a corolla limb. In contrast, *A. lycioides* is a shrub, producing axillary solitary flowers that typically lack a corolla limb but are sometimes found with a very small corolla limb (Figs. 1E, 1F). In terms of floral color (Table 1), *A. versicolor* has a whitish corolla tube with a purple corolla limb, *A. mekongensis* has a pure white corolla tube and limb with small brownish dots scattered across the flower, and *A. lycioides* has a waxy greenish-white tube with a dense concentration of dark purple dots inside the corolla (Fig. 1). The flowering periods of both *A. versicolor* and *A. mekongensis* occur from August to December and fruits are mature approximately 10–12 weeks later. The flowering period of *A. lycioides* is shorter, from early September until late October, and fruits are mature approximately 10–12 weeks later (Table 1).

All three sympatric species have a campanulate floral shape but differ in size (Table 1, Figs. 1 and 2). ANOVA results revealed that these species are significantly different in terms of corolla length ($F_{2,12} = 40$, $p < 0.001$), corolla tube diameter ($F_{2,12} = 11.58$, $p < 0.01$), stamen length ($F_{2,13} = 71.51$, $p < 0.001$), and pistil length ($F_{2,11} = 116.2$, $p < 0.001$) (Fig. 2). These floral characters appear to be smallest in *A. lycioides* and largest in *A. versicolor*, while *A. mekongensis* is intermediate for all studied characters. The corolla length of *A. lycioides* is significantly shorter than that of *A. versicolor* and *A. mekongensis*, although the latter two are not significantly different. The corolla tube diameter of *A. mekongensis* is not significantly different from other two species, while the diameters of *A. lycioides* and *A. versicolor* are significantly different. In terms of stamen and pistil lengths, all three species are significantly different from each other (Fig. 2).

### Floral anthesis and visitor observation

The floral anthesis of both *A. mekongensis* and *A. versicolor* were obtained from *Jirabanjongjit et al. (2024)* (see Table 1). The flowers of *A. lycioides* generally start to open around 5.00 h, are fully open around 8.00−9.00 h, and last until the evening of the following day (Table 1). The longevity of *A. mekongensis* and *A. lycioides* flowers is approximately 36–40 h, while that of *A. versicolor* is 15–16 h.

The flowers of *A. versicolor* and *A. mekongensis* are both pollinated by *Xylocopa latipes* and *X. aestuans* (*Jirabanjongjit et al., 2024*) (Figs. 3A–3D), and are also visited by several diurnal florivores, visitors, and nectar robbers (Figs. 3E–3G). Several diurnal animal taxa were observed visiting *A. lycioides* flowers, including an unknown bee (Anthophila; Fig. 3H), wasps (Vespidae; Fig. 3I), several ants (Formicidae), *Cinnyris jugularis* sunbirds (Nectariniidae; Fig. 3J), cockroaches (Blattodea), and skipper butterflies (Hesperiidae). No nocturnal visitors were observed. While some differences in species richness and abundance were observed between the two study years, visitation rates of animal taxa were not significantly different (Fig. 4). During the 2019 flowering season we observed only two visitor taxa (Formicidae and the unknown bee species), both of which were uncommon (Fig. 4A). During the 2020 flowering season we observed five visitor taxa (Fig. 4B). Vespid wasps were the most frequent visitors (Fig. 4B). They entered the corolla to

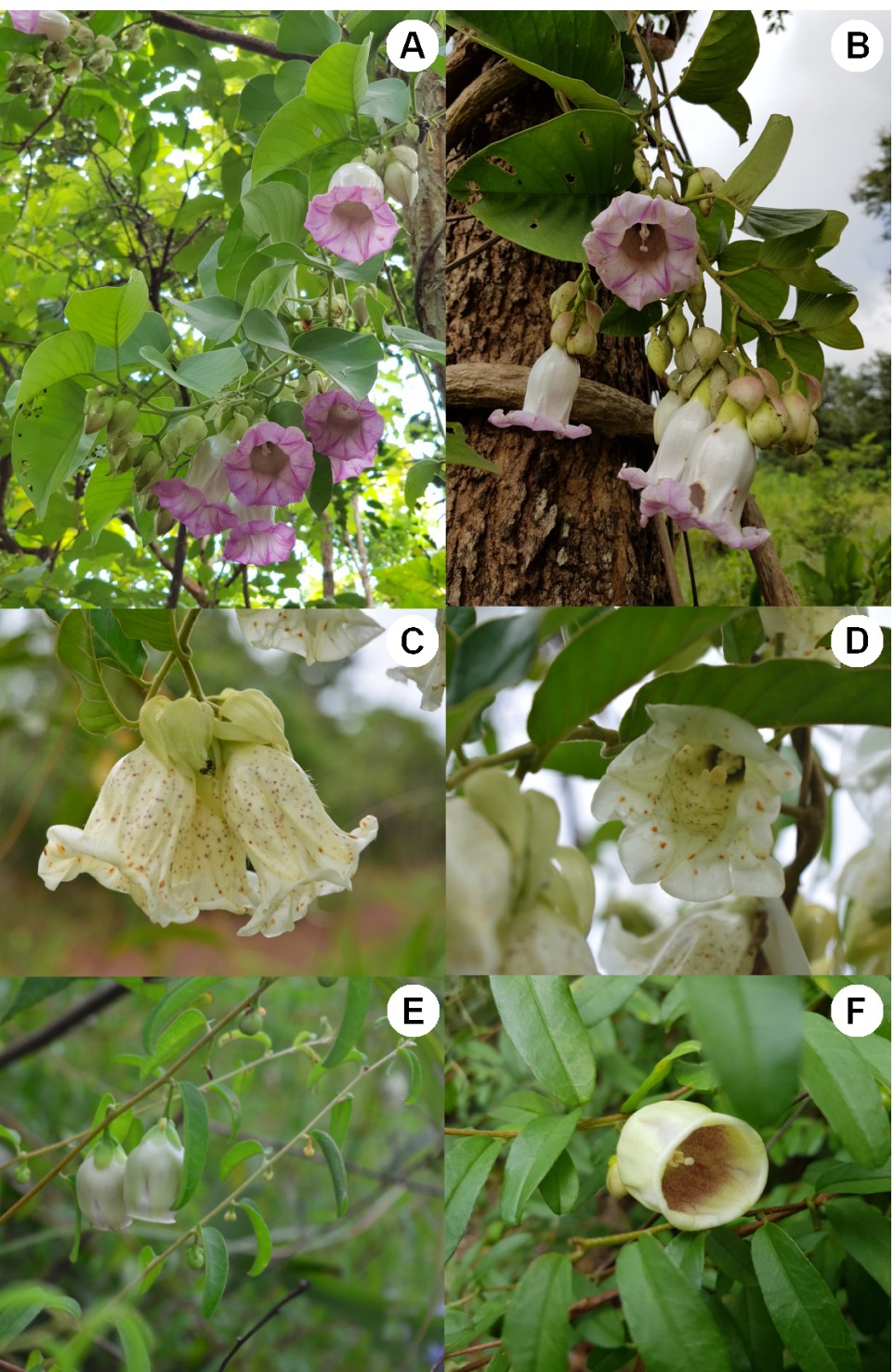

**Figure 1** **Photos showing the floral characters of three sympatric *Argyreia* species.** (A, B) *A. versicolor*, (C, D) *A. mekongensis*, and (E, F) *A. lycioides*. Photos A, B and F were taken by Yotsawate Sirichamon, and C, D and E were taken by Tripatchara Atiratana.

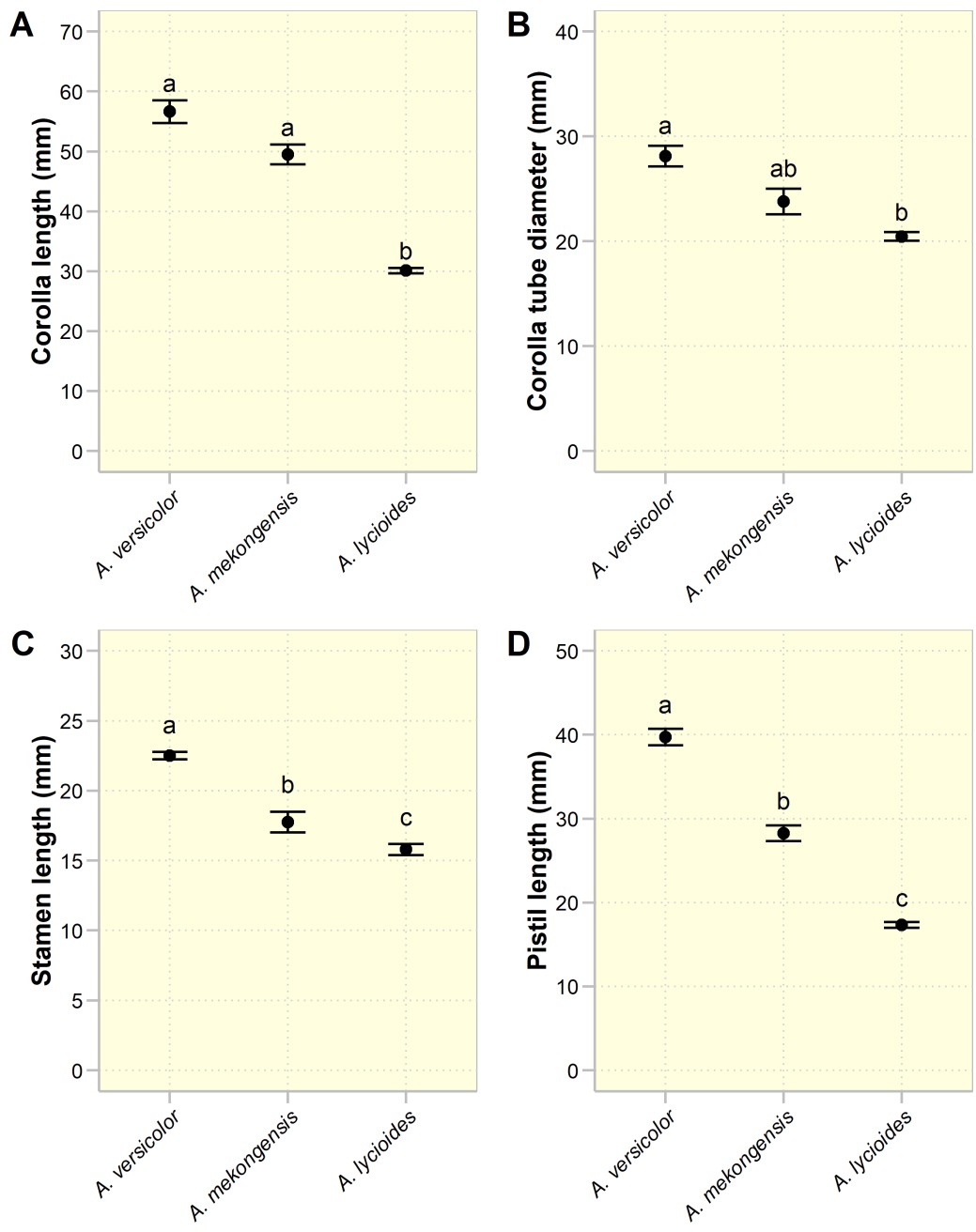

**Figure 2 A comparison of quantitative floral characters in three sympatric *Argyreia* species.** (A) Corolla length. (B) Corolla tube diameter. (C) Stamen length. (D) Pistil length. Circles and error bars denote means and standard errors. Species with different lowercase letters are significantly different ($p < 0.05$).

forage on nectar, during which their thorax was observed to touch the anthers and stigmas; pollen was observed on their thorax (Fig. 3I). The unknown bee species was also observed contacting floral reproductive structures, but was only rarely recorded visiting flowers. The cockroaches and skipper butterflies visited flowers several times (Fig. 4B) but appear

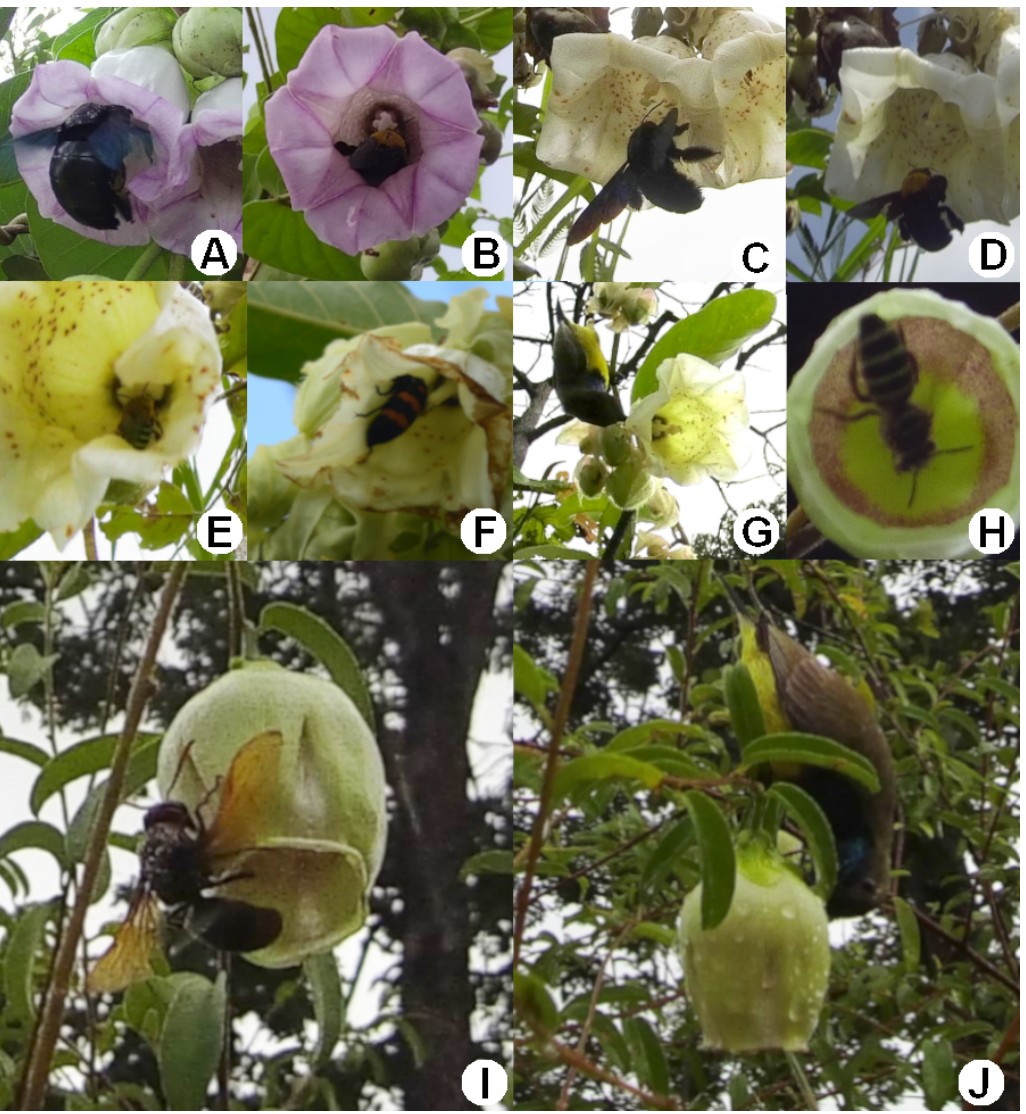

**Figure 3** **Animal visitors of three sympatric species of *Argyreia*.** (A) *Xylocopa latipes* visiting *A. versicolor*. (B) *X. aestuans* visiting *A. versicolor*. (C) *X. latipes* visiting *A. mekongensis*. (D) *X. aestuans* visiting *A. mekongensis*. (E) *Amegilla* sp. visiting *A. mekongensis*. (F) *Mylabris phalerata* beetle consuming the corolla of *A. mekongensis*. (G) *Cinnyris jugularis* sunbird robbing nectar from *A. mekongensis*. (H) Unknown bee species (Anthophila) visiting *A. lycioides*. (I) Unknown wasp species (Vespidae) visiting *A. lycioides*. (J) *Cinnyris jugularis* sunbird robbing nectar from *A. lycioides*. Photos credited to Awapa Jirabanjongjit.

not to pollinate flowers, as they simply crawled along the outside of the corolla. Ants were occasionally observed walking on the inside of the corolla, in addition to the outside, but were never observed contacting anthers or stigmas. Sunbirds were relative frequent visitors in 2020 (Fig. 4B), but were observed robbing nectar by perching on branches near flowers and piercing the corolla base with their beaks (Fig. 3J).

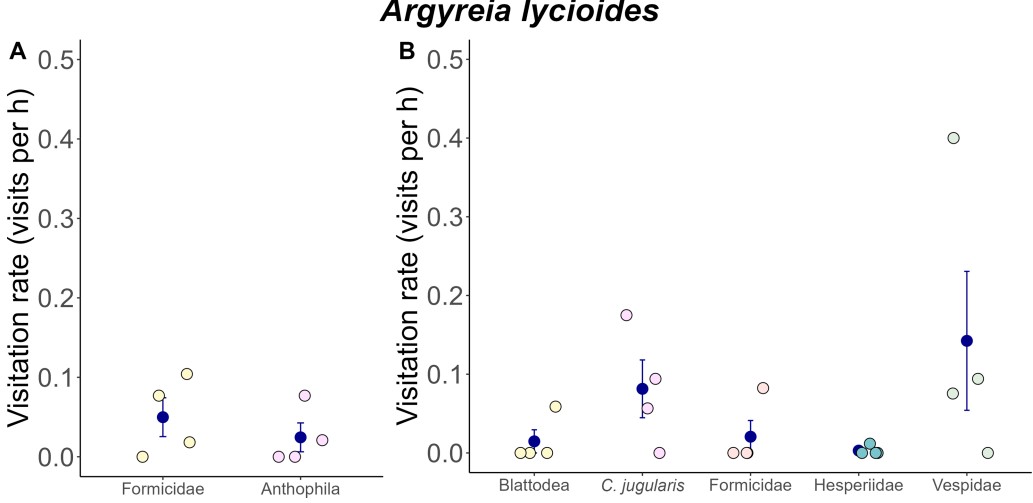

**Figure 4** **Visitation rates of animal visitors observed at *A. lycioides* flowers.** (A) 2019 ($n = 4$ plants) and in (B) 2020 ($n = 4$ plants). The blue circles and error bar denote means and standard errors, while pastel-colored jittered points show the distributions of the raw data. Note: Only Anthophila and Vespidae contacted floral stigmas and anthers, and are the potential pollinators of *A. lycioides*.

## Histochemistry of floral nectary discs

The sectioned nectaries of the three study species revealed some similarities and some differences among the tested compounds (Table 2). NADI reagent tested positive in all three species, revealing that terpenes are produced and/or accumulated in the epidermis, around the nectary ducts, and in the parenchyma cells of *A. versicolor* (Fig. 5A) and *A. mekongensis* (Fig. 6A), while terpenes were broadly detected throughout the entirety of the nectary disc of *A. lycioides* in comparison to other two species (Fig. 7A). Similarly, flavonoids were detected in all species throughout the nectary disc (Figs. 5D, 6G, 7C). In contrast, lipids were found only in *A. mekongensis*, as detected by both Sudan Black B and Sudan III, appearing to accumulate in the epidermis layer and nectary ducts (Figs. 6C, 6E), while the other two study species tested negative for lipids using both Sudan Black B and Sudan III.

## Morphology and histochemistry of staminal trichomes

*Argyreia versicolor* has two types of trichomes (Table 2), glandular trichomes (Figs. 5B, 5C, 5E) and non-glandular trichomes (Fig. 5F). The trichomes were dispersed across the lower part of the filaments, and the highest density of trichomes and the longest trichomes were found at the center of the distribution. Glandular trichomes were shorter than non-glandular trichomes and were fewer in number and shorter in length towards the margins of their distribution. Each glandular trichome consisted of a head cell (apical cell), stalk, and basal cell; the head cells are unicellular apical glands which are either rounded cylindrical, obovoid (Fig. 5C), or globose (Fig. 5E). Stalks were observed to have different lengths; long stalks (Fig. 5C) were mostly found at the very base of the filaments while short stalks (Fig. 5E) were densely scattered around the middle of filaments and sparsely

**Table 2  Results of histochemical analysis testing for the presence of terpenes, lipids, and flavonoids in the floral nectary and staminal trichomes of three sympatric *Argyreia* species.**

| Floral organ | Species | Reagent tested | | | |
|---|---|---|---|---|---|
| | | NADI (terpenes) | Sudan III (lipids) | Sudan Black B (lipids) | Naturstoff (flavonoids) |
| Nectary discs | *A. versicolor* | √ | – | – | √ |
| | *A. mekongensis* | √ | √ | √ | √ |
| | *A. lycioides* | √ | – | – | √ |
| Glandular staminal trichomes | *A. versicolor* | √ | – | – | – |
| | *A. mekongensis* | √ | √ | √ | – |
| | *A. lycioides* | √ | – | – | – |
| Non-glandular staminal trichomes | *A. versicolor* | – | – | – | √ |
| | *A. mekongensis* | N/A | N/A | N/A | N/A |
| | *A. lycioides* | N/A | N/A | N/A | N/A |

**Notes.**
√, positive result;  –,  negative result;  N/A,  not applicable.
(*A. mekongensis* and *A. lycioides* do not possess non-glandular staminal trichomes).

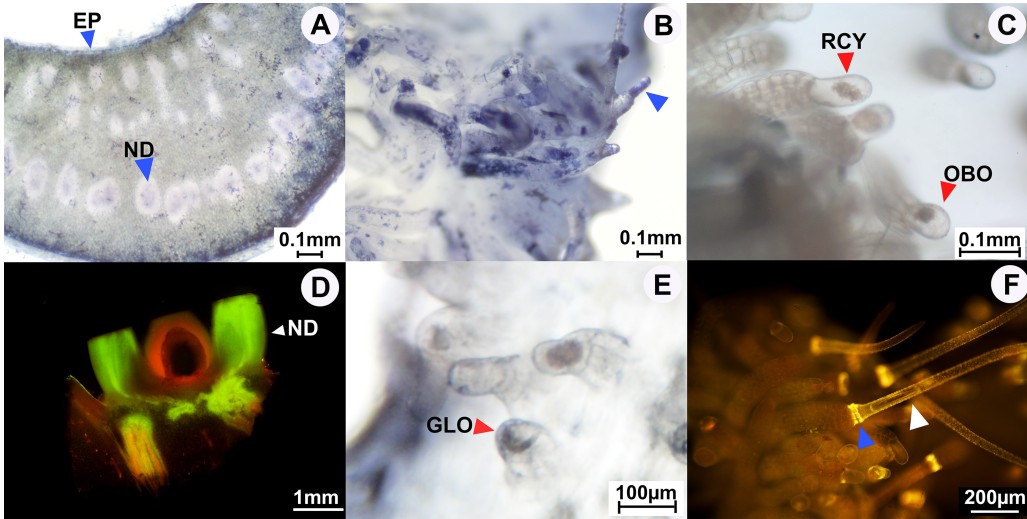

**Figure 5  Results of histochemical analysis conducted in *A. versicolor*.** (A) Transversal section of the floral nectary showing the presence of terpenes; positive staining shown at the blue arrow pointing to a nectary duct (ND) and the nectary epidermis (EP). (B) Staminal trichomes showing the presence of terpenes; positive staining shown by the blue arrow pointing to the apical gland cell. (C) Unstained long glandular trichomes at the base of staminal filaments; red arrows pointing to rounded cylindrical (RCY) and obovoid (OBO) apical gland cells. (D) Longitudinal section of the floral nectary showing the presence of flavonoids under a fluorescence microscope; positive staining shown by the white arrow pointing to the nectary disc. (E) Unstained short glandular trichomes at the middle of staminal filaments; red arrow pointing to an apical gland cell that is globose shaped (GLO). (F) Staminal trichomes stained with Naturstoff reagent and viewed under fluorescence microscope reveal the presence of flavonoids; strong staining shown at the blue arrow, white arrow pointing to an apical cell of a non-glandular trichome. Photos credited to Awapa Jirabanjongjit.

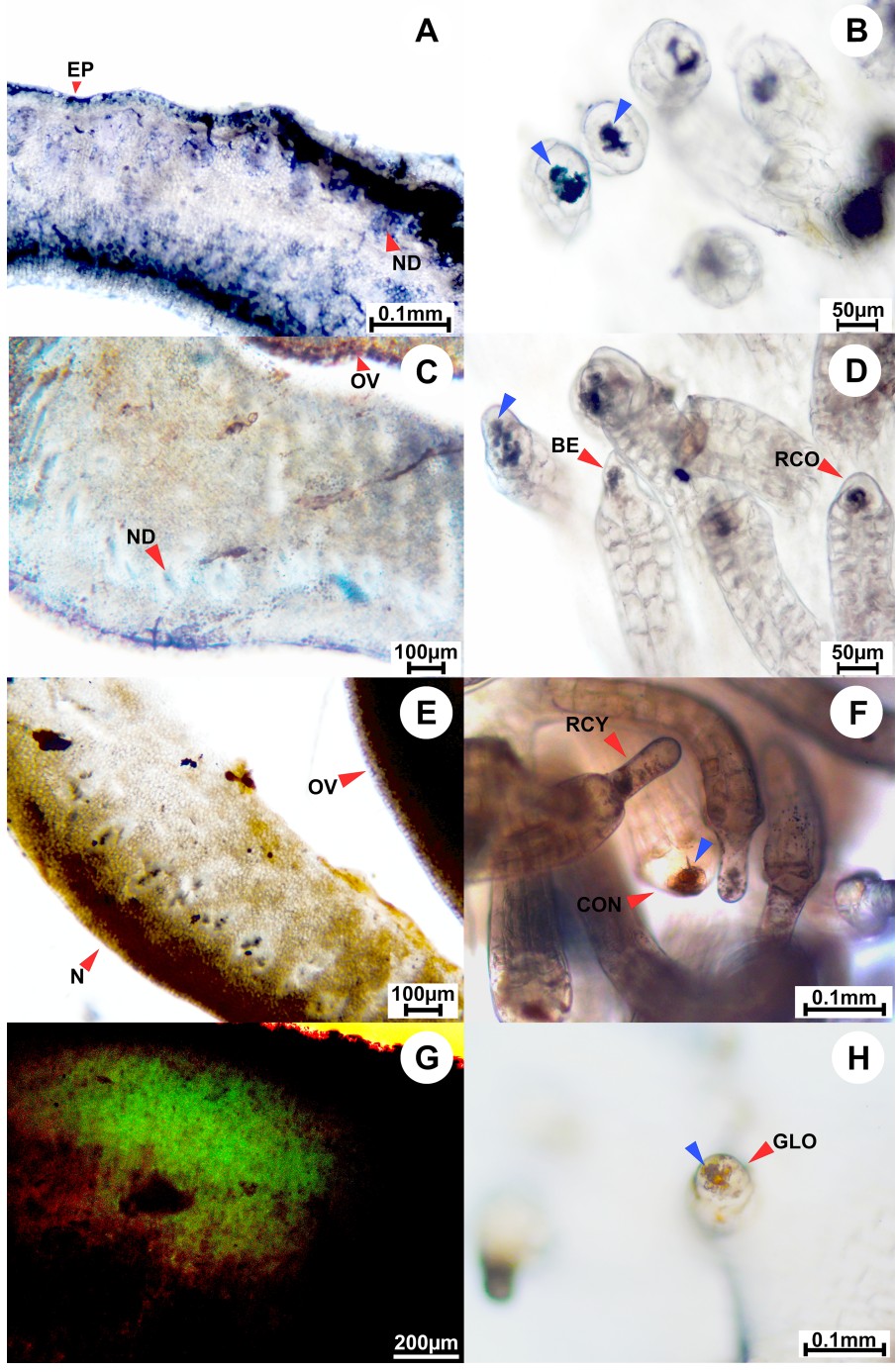

**Figure 6  Results of histochemical analysis conducted in *A. mekongensis*.** (A) Transversal section of the floral nectary showing the presence of terpenes; positive staining shown at the red arrows pointing to the nectary duct (ND) and nectary epidermis (EP). (B) Staminal trichomes tested positive for terpenes; blue arrows point to apical gland cells showing terpenes inside of the glands. (C) Transversal section of the floral nectary stained with Sudan Black B showing the presence of lipids; (continued on next page...)

**Figure 6 (…continued)**
positive staining shown at the red arrow pointing to the nectary duct (ND). (D) Staminal trichomes
stained with Sudan Black B tested positive for lipids; red arrows point to apical gland cells containing
lipids, and demonstrate the different types of gland cells: BE, bell-shaped; RCO, rounded conical. (E)
Transversal section of the floral nectary stained with Sudan III showing the presence of lipids in black. (G)
Transversal section of the floral nectary stained with Naturstoff reagent and viewed under a fluorescence
microscope showing the presence of flavonoids. (F, H) Staminal trichomes stained with Sudan III
tested positive for lipids; blue arrows point to lipids inside of the apical gland cells, while red arrows
demonstrate the different types of gland cells: RCY, rounded cylindrical; CON, convex; GLO, globose.
Other abbreviations: N, nectary; OV, ovary; EP, epidermis. Photos credited to Awapa Jirabanjongjit.

scattered near the tops of filaments. The average density of glandular trichomes was 36.1
$\pm$ 2.3 trichomes per mm$^2$. We also observed simple non-glandular staminal trichomes
(Fig. 5F), which were present only in this species and only at the center of the filament base,
with an average density of 5.1 $\pm$ 0.5 trichomes per mm$^2$. The non-glandular trichomes
consisted of a basal cell and a long slender apical cell. The glandular staminal trichomes
tested positive for terpenes (Fig. 5B) while the simple non-glandular staminal trichomes did
not (Table 2). Both glandular and non-glandular trichomes tested negative for lipids (Table
2). Flavonoids were detected in the apical cells of non-glandular trichomes, especially where
the apical cell connects to the base (Fig. 5F), but not in glandular trichomes (Table 2).

Only glandular trichomes were observed for *A. mekongensis*, and they were distributed
across the base of filaments, densely at the center of their distribution and more sparsely
towards the margins of their distribution, with an average density of 32.7 $\pm$ 5.6 trichomes
per mm$^2$. These glandular trichomes also consisted of a head cell (apical cell), stalk, and
basal cell. The head cells are unicellular apical glands and five gland shapes were observed;
rounded conical (Fig. 6D), bell-shaped (Fig. 6D), rounded cylindrical (Fig. 6F), convex
(Fig. 6F), and globose (Fig. 6H). Stalks were longer at the center of their distribution
and shorter towards the margins of their distribution. Histochemical analysis revealed
the presence of terpenes (Fig. 6B) and lipids (Figs. 6D, 6F, 6H), both of which appear to
accumulate in the glands (Table 2). However, flavonoids were not detected (Table 2).

We also only observed glandular trichomes in *A. lycioides*, which were distributed across
the base of the filaments, densely at the center of their distribution and more sparsely
towards the margins of their distribution, with an average density of 27.0 $\pm$ 4.0 trichomes
per mm$^2$. Similar to the other two study species, these glandular trichomes consisted of
a head cell (apical cell), stalk, and basal cell. The apical gland cells were observed to have
four shape types: rounded cylindrical (Fig. 7B), globose (Fig. 7B), obovoid (Fig. 7D), and
pyriform (Fig. 7D). In contrast to the other two study species, the glandular trichomes of
*A. lycioides* have very short unicellular stalks (Fig. 7D). Histochemical analysis revealed the
presence of terpenes (Fig. 7B), while lipids and flavonoids were not detected (Table 2).

## DISCUSSION

### Floral visitors in relation to floral characters

We observed high overlap in floral visitor composition between *A. versicolor* and *A.
mekongensis*, and some overlap between *A. mekongensis* and *A. lycioides*, but no overlap in
the taxa visiting *A. versicolor* and *A. lycioides*. *Argyreia versicolor* and *A. mekongensis* were

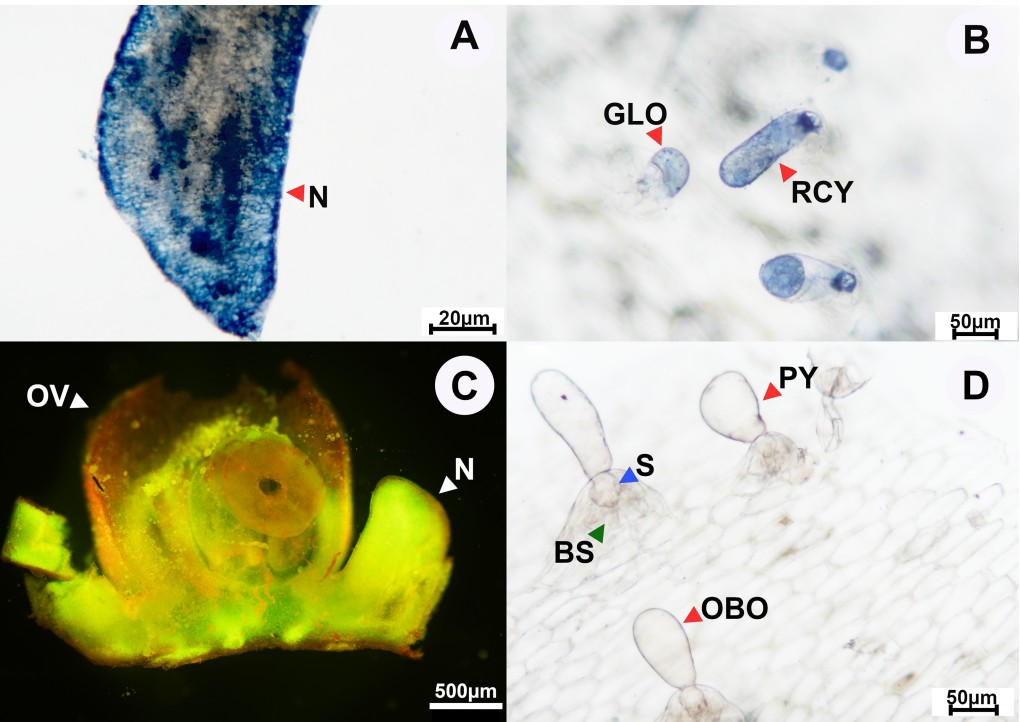

**Figure 7** **Results of histochemical analysis conducted in *A. lycioides*.** (A) Longitudinal section of the floral nectary showing the presence of terpenes. (B) Staminal trichomes tested positive for terpenes; positive staining shown at the red arrows, which also demonstrate some of the different types of apical gland cells: GLO, globose; RCY, rounded cylindrical. (C) Longitudinal section of the floral nectary stained with Naturstoff reagent and viewed under a fluorescence microscope showing the presence of flavonoids. (D) Unstained glandular trichomes; red arrows demonstrate some of the different types of apical gland cells: PY, pyriform; OBO, obovoid. Other abbreviations: N, nectary; OV, ovary; BS, basal cells; S, stalk. Photos credited to Awapa Jirabanjongjit.

both almost exclusively visited by *Xylocopa* carpenter bees (*X. aestuans* and *X. latipes*). These species are very similar in terms of corolla shape and size, both of which allow their large bee pollinators to enter the flower and contact floral reproductive structures with their thorax. Moreover, these species potentially attract pollinators with their relatively showy floral displays: bright colors, a large corolla limb (*Jirabanjongjit et al., 2024*), and numerous flowers per inflorescence (5–9 flowers). While these two species do differ in color (*A. versicolor* is purple and white, while *A. mekongensis* is pure white), *Xylocopa* bees have been reported to favor both purplish-white and creamy white flowers (*Raju & Rao, 2006*).

Interestingly, the flowers of *A. mekongensis* are potentially capable of being pollinated over two days (although stigma receptivity still needs to be tested), while those *A. versicolor* open for only a single day. The long lifespan of *A. mekongensis* flowers is unusual in the Convolvulaceae, as most morning glory flowers bloom for no more than 12 h (*Hassa, Traiperm & Stewart, 2023*). Such differences in floral longevity suggest that *A. mekongensis* flowers possibly receive twice as many visits as *A. versicolor* flowers, given that carpenter

bee visitation rates to the two species are comparable (*Jirabanjongjit et al., 2024*). However, these differences do not appear to have noticeable effects on plant reproduction, as naturally pollinated flowers of the two *Argyreia* species set fruit that are similar in terms of both fruit weight and seed number (*Jirabanjongjit et al., 2024*).

We also observed some overlap in the animal taxa that visited *A. mekongensis* and *A. lycioides*, namely, wasps, medium-sized bees, skipper butterflies, and sunbirds. Of these taxa, only wasps in the family Vespidae were frequent visitors to and likely pollinators of *A. lycioides* (see paragraph below). The remaining taxa were uncommon visitors and unlikely pollinators (Fig. 4). For example, for both *Argyreia* species, skipper butterflies (Hesperiidae) were only ever observed on the outside of the corolla, and sunbirds (*Cinnyris jugularis*) were only ever observed robbing nectar. The presence of sunbirds at these *Argyreia* species is somewhat surprising given that we observed only trace amounts of sticky nectar at the base of the corolla tube (and their pendant orientation likely does not allow large quantities of nectar to accumulate) and most flowers attractive to birds produce copious amounts of dilute nectar (*Stiles, 1981*; *Johnson & Nicolson, 2008*). Moreover, *A. mekongensis* and *A. lycioides* flowers have relatively pale and dull colors in contrast to most bird-visited flowers (*Stiles, 1981*). Thus, additional research is needed to understand the behavior of sunbirds at *A. mekongensis* and *A. lycioides*. The similarities in floral visitor composition between *A. mekongensis* and *A. lycioides* may be due to their similar coloration and floral heights. Both species have pale-colored flowers and were found about 1.5 m above ground, whereas twining *A. versicolor* was usually found climbing tall trees and its flowers were typically around 5 m above ground.

*Argyreia lycioides* is the most distinct in shape and size among the three study species. Its flowers are significantly smaller than those of the other two *Argyreia* species, and it also appears to attract smaller pollinators compared to the other two species. The flowers of *A. mekongensis* and *A. versicolor* were pollinated by *Xylocopa* carpenter bees (*Jirabanjongjit et al., 2024*), while *A. lycioides* appears to be pollinated primarily by wasps in the family Vespidae. Key floral features that distinguish *A. lycioides* from the other two species are its floral arrangement (axillary solitary instead of inflorescence), flower size, and reduced corolla limb (minimal to absent). The flowers of *A. lycioides* are too small for carpenter bees, but are appropriately sized for wasps or smaller bees, as were observed in this study. Wasps appear to be the main pollinators of *A. lycioides* given the frequency of their visits and their consistent contact with stigmas and anthers. Wasps can be found as pollinators of both generalist and specialized flowers (*Heithaus, 1979*; *Nilsson, 1981*; *Kephart, 1983*; *Vieira & Shepherd, 1999*; *Ollerton et al., 2003*; *Johnson, 2005*; *Shuttleworth & Johnson, 2006*; *Shuttleworth & Johnson, 2009*; *Johnson, Ellis & Dötterl, 2007*). While *Faegri & Van der Pijl (1979)* did not specifically describe a wasp pollination syndrome, evidence suggests that wasps often pollinate easily approachable flowers that have dull or cryptic coloration, a strong or unusual scent, and concentrated nectar (*Heithaus, 1979*; *Proctor, Yeo & Lack, 1996*; *Ollerton & Watts, 2000*; *Johnson, Ellis & Dötterl, 2007*; *Shuttleworth & Johnson, 2009*). According to *Kingston & Mc Quillan (2000)*, flowers that are visited by wasps mainly have pale colors, followed by yellow and some purple flowers. These results correspond with our findings, as *A. lycioides* has a greenish white corolla with a dense concentration of

dark purple dots on the inside the corolla, and the wide corolla entrance and tube make nectar easily accessible. However, given that our study only opportunistically examined a single population, additional research is needed to assess the visitation frequency and effectiveness of animal taxa visiting *A. lycioides* to determine its main pollinators.

## Histochemistry of the floral nectary discs

In all three study species, the floral nectary surrounds the base of the ovary; this nectary disc is a conserved character within the Convolvulaceae (*Govil, 1972*; *Deroin, 1992*; *Galetto & Bernardello, 2004*; *Wright, Welsh & Costea, 2011*). However, investigation of terpenes, flavonoids, and lipids in the floral nectaries revealed some differences between the three sympatric *Argyreia* species.

Terpenes were detected in the floral nectaries of all three study species, and have also been reported in several other plant species (*Giuliani, Bini & Lippi, 2012*; *Machado & Souza, 2016*; *Wiese et al., 2018*; *Farinasso et al., 2021*; *Jirabanjongjit et al., 2021*; *Chitchak, Stewart & Traiperm, 2022*). Terpenes are important secondary metabolites in plants that contribute to pollinator attraction by providing scent compounds (*Knudsen & Gershenzon, 2006*) that are recognized to attract bees (*Bergström, Dobson & Groth, 1995*; *Robertson et al., 1995*). Moreover, terpenes can contribute to plant-insect interactions for bees that forage for biologically active plant products (*Harrewijn, Minks & Mollema, 1994*; *Stevenson, Nicolson & Wright, 2017*). Therefore, apart from the food resources provided by nectar, the floral nectary can also produce other important chemical substances to attract and reward pollinators.

Flavonoids were also detected in the floral nectaries of all three study species, are prevalent throughout plants and their tissues especially in higher plants (*Wollenweber & Dietz, 1981*; *Harborne, 1988*; *Taylor & Grotewold, 2005*), and have previously been reported in several plant taxa (*Ferreres et al., 1996*; *Truchado et al., 2008*; *Machado & Souza, 2016*; *Jirabanjongjit et al., 2021*; *Chitchak, Stewart & Traiperm, 2022*). Several functions of flavonoids are well-known, such as their role in plant reproduction, namely, as a color attractant that advertises flowers to pollinators and fruits to seed dispersers (*Dakora, 1995*). Flavonoids can also absorb UV wavelengths, providing visual cues that guide bees or other insects to floral nectar (*Thorp et al., 1975*; *Harborne, 1979*; *Agati & Tattini, 2010*). Additionally, flavonoids can protect nectar from pathogens or microbes, preserving it for pollinators, which can also benefit plant reproduction (*Treutter, 2005*).

In contrast to terpenes and flavonoids, lipids were only found in one of the three study species, *A. mekongensis*. Lipids have been reported in the floral nectaries of many plant taxa, such as Anacardiaceae, Bignoniaceae, Convolvulaceae, and Orchidaceae (*Figueiredo & Pais, 1992*; *Stpiczyńska, 1997*; *Stpiczyńska & Matusiewicz, 2001*; *Stpiczynska & Davies, 2006*; *Kowalkowska et al., 2015*; *Machado & Souza, 2016*; *Tölke et al., 2018*; *Phukela, Adit & Tandon, 2021*; *Jirabanjongjit et al., 2021*; *Chitchak, Stewart & Traiperm, 2022*). Lipids are frequently found in the nectar of vertebrate-pollinated species due to their importance in the diets of vertebrates (*Varassin, Trigo & Sazima, 2001*; *Gumede & Downs, 2020*), however, their potential role in plant-insect interactions has not been widely studied. Previous research suggests that the positive detection of lipids in the floral nectary could indicate

the presence of laticifers (*Martins et al., 2012*), or may provide nutrition for nectar-feeding insects, in addition to polysaccharides (*Bernardello, 2007*). It is noteworthy that lipids were found only in *A. mekongensis*, but not the other two *Argyreia* species. These lipids did not appear to attract different pollinators from the other two congeners, and floral visitation rates were similar across all three study species. Thus, further work is needed to elucidate the role of lipids in the floral nectary.

## Histochemistry of the staminal trichomes

*Argyreia versicolor* is the only species examined in this study that has both non-glandular and glandular trichomes on the staminal filaments. Non-glandular staminal trichomes appear to be very uncommon, as *Chitchak, Stewart & Traiperm (2024)* observed the staminal trichomes of 73 taxa and observed non-glandular trichomes in *A. versicolor*. Traditionally, non-glandular trichomes have been considered unimportant in the storage, production, and secretion of biologically active compounds (*Werker, 2000*). However, non-glandular trichomes have been found to store phenolic compounds, despite not having secretion abilities, and such phenolics are important in the protection against and regulation of biotic and abiotic stresses (*Koudounas et al., 2015*; *Karabourniotis et al., 2020*). Among secondary metabolic compounds, flavonoids, which are phenolic compounds, have been shown to substantially accumulate in non-glandular trichomes (*Skaltsa et al., 1994*; *Valkama et al., 2004*; *Tattini et al., 2007*; *Koudounas et al., 2015*), as we also observed in *A. versicolor*. Non-glandular trichomes are typically considered to provide physical plant defenses against biotic or abiotic stresses, such as protection against insect oviposition or herbivory (*Levin, 1973*; *Baur, Binder & Benz, 1991*), or as protection against drought (*Ichie et al., 2016*), low or high humidity, high solar radiation, or high light intensity (*Werker, 2000*; *Ichie et al., 2016*). Thus, it is possible that the non-glandular staminal trichomes found in *A. versicolor* may help protect the ovary from herbivory, as the ovary is located directly below the staminal trichomes.

Glandular trichomes were observed in all three study species, and had similar distribution patterns, but varied in some of their morphological features. *Argyreia lycioides* had shorter stalks than the other two species, which may be due to phenotypic integration of floral traits such as corolla size, as was reported in *Chitchak, Stewart & Traiperm (2024)*. The function of staminal trichomes is still unclear and they have received less attention than other types of trichomes. However, a recent study by *Dieringer & Cabrera (2022)* suggested one ecological advantage of staminal trichomes in *Agalinis auriculata*, in which they appear to facilitate the grasping of flowers during buzz pollination; filament trichomes had a positive effect on pollen removal in sternotribic pollination but a negative effect in nototribic pollination. *Riviere et al. (2013)* reported a different function of filament trichomes in the genus *Cuscuta*, where glandular trichomes appear to play a role in the protection of the nectar or ovary (ovules).

Glandular trichomes usually secrete and accumulate specific secondary metabolites such as terpenes and other essential oils (*Metcalf & Kogan, 1987*). In the species that we examined, there were some differences in the histochemical results of glandular staminal trichomes, but terpenes were found in all species. In general, terpenes are reserved in

specialized structures within plant tissues, such as secretory cavities, resin canals, latex canals, and glandular trichomes (*Holopainen et al., 2013*). Previous studies examining terpenes have mostly reported their presence in the glandular trichomes of leaves, and they are generally recognized as being defensive substances against microbes, fungi and/or herbivores (*Harborne, 1993*; *Kelsey, Reynolds & Rodriguez, 1984*; *Olaranont et al., 2018*). However, more recent work has started to improve our understanding of how insects respond to terpenes and their function in attracting pollinators (*Raguso & Light, 1998*; *Dudareva et al., 2006*; *Knudsen & Gershenzon, 2006*). In contrast to the ubiquity of terpenes, lipids were only detected in the glandular staminal trichomes of *A. mekongensis* while flavonoids were not detected in the glandular staminal trichomes of any species. Previous work examining other morning glory species, such as *Rivea ornata* (*Chitchak, Stewart & Traiperm, 2022*) and *Argyreia siamensis* (*Jirabanjongjit et al., 2021*), have hypothesized that staminal trichomes are important for pollinator attraction since glandular staminal trichomes contain chemical substances such terpenes and flavonoids. The results of this study indicate that glandular staminal trichomes, with their accumulation of terpenes, may help protect the nectar and ovary from microbes, fungi, and herbivores, and/or may help attract pollinators and guide them to the nectar.

## CONCLUSIONS

The findings of this study reveal that the three sympatric, co-flowering *Argyreia* species exhibit some similarities and some differences in terms of floral characters and histochemical compounds, which appear to influence their interactions with floral visitors. Notably, the two twining species, *A. versicolor* and *A. mekongensis*, have similar floral shapes and sizes, and are both pollinated by large carpenter bees. In contrast, *A. lycioides* is a perennial shrub with a smaller corolla tube, and appears to be pollinated by wasps (Vespidae) and possibly small- to medium-sized bees. All species exhibited trichomes at the base of staminal filaments; glandular trichomes were observed in all species but non-glandular trichomes were found only in *A. versicolor*. Moreover, because the glandular trichomes of all species contained volatile terpenes, these structures and chemical compounds may help attract pollinators and/or guide them to the base of the corolla, where pollinators were observed foraging. In contrast, non-glandular trichomes are typically associated with herbivore defense, which may help explain why florivores were not observed on *A. versicolor* even though they were observed on sympatric congeners. Furthermore, terpenes and flavonoids found in the floral nectary of all three study species may contribute to pollinator attraction while lipids in the nectar (found only in *A. mekongensis*) may provide additional nutrition for pollinators. Overall, floral morphological differences contribute to some pollinator partitioning, with *A. lycioides* utilizing unique pollinators, but *A. versicolor* and *A. mekongensis* share the same pollinators. Additional research is needed to determine whether these congeners rely on other mechanisms to reduce competition for pollination (*e.g.*, pollinator constancy or differential pollen placement), or whether competition and interspecific pollen transfer reduce their reproductive success. The knowledge gained from this study regarding flower morphology and chemical compounds, in combination

with what we know about the breeding system of the rare species, *A. versicolor* and *A. mekongensis* (*Jirabanjongjit et al., 2024*), are necessary for plant conservation in terms of protecting their shared pollinators.

## ACKNOWLEDGEMENTS

We would like to thank everyone who helped with field work, including Dr. Pongsakorn Kochaipat, Poomphat Srisombat, Tripatchara Atiratana, Kanjana Pramali, Dr. Yanisa Olaranont, and Piriya Hassa. We are also grateful to Dr. Natapot Warrit for assisting with insect identification. We thank Prof. Dr. Curtis Daehler, Dr. Jose G Garcia-Franco and two anonymous reviewers for their helpful comments on an earlier version of this manuscript.

### Funding

This research was funded by Mahidol University (MU's Strategic Research Fund awarded to Paweena Traiperm and Alyssa B. Stewart). The funders had no role in study design, data collection and analysis, decision to publish, or preparation of the manuscript.

### Grant Disclosures

The following grant information was disclosed by the authors:
Mahidol University.

### Competing Interests

The authors declare there are no competing interests.

### Author Contributions

- Awapa Jirabanjongjit conceived and designed the experiments, performed the experiments, analyzed the data, prepared figures and/or tables, authored or reviewed drafts of the article, and approved the final draft.
- Alyssa B. Stewart conceived and designed the experiments, performed the experiments, analyzed the data, prepared figures and/or tables, authored or reviewed drafts of the article, and approved the final draft.
- Natthaphong Chitchak conceived and designed the experiments, performed the experiments, analyzed the data, prepared figures and/or tables, authored or reviewed drafts of the article, and approved the final draft.
- Chakkrapong Rattamanee performed the experiments, authored or reviewed drafts of the article, and approved the final draft.
- Paweena Traiperm conceived and designed the experiments, performed the experiments, analyzed the data, prepared figures and/or tables, authored or reviewed drafts of the article, and approved the final draft.

### Data Availability

The raw data are available in the Supplementary File.

## Supplemental Information

Supplemental information for this article can be found online at http://dx.doi.org/10.7717/peerj.17866#supplemental-information.

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
