# Peer review of "Variation in floral morphology, histochemistry, and floral visitors of three sympatric morning glory species"

_PeerJ, doi:10.7717/peerj.17866_

## Round 0.1 · original submission · Major Revisions

The reviewers, particularly reviewers 1 and 2 requested major revision, including more information on the methods / experimental design / sampling and statistical analyses, and inclusion of quantitative data wherever possible. A clearer explanation of what work has been previously been done is needed, along with a statement of the knowledge gap that this study seeks to fill. If you feel that you can address the reviewers' concerns, you may wish to submit a revised manuscript.

Reviewer 1 ·

Basic reporting

I now have read the ms “Variation in floral morphology, histochemistry and floral visitors of three sympatric morning glory species” that aiming to link floral visitors with variation of floral traits. I have major concerns about its quality.
Introduction
I find introduction somewhat weak in justification. I suggest that you should expand upon the knowledge gap being filled.

Experimental design

The authors report that they conducted observations about floral visitors, but there are some aspects in the design and data collection that is not mentioned, for example, what time were the observations made?, How many flowers were observed?, How the date were analyzed ? How many hours in total of observation? How many days?, and statistical analysis should be included.
I am also concerned about the morphology floral data collection, the authors not mentioned which floral traits were considered in the study, and the authors not explain how analyzed the date.

Validity of the findings

The interpretation of results is too speculative, the current version does not explore causal mechanisms of the interation floral traits and floral visitors.

Therefore, I suggest to the Editors-in-Chief to reject the current version of
the manuscript.

·

Basic reporting

Ms presents a study describing the relationship between the morphology of flowers, some of their histochemical characteristics, and flower visitors in three sympatric species of the genus Argyreira. The plants studied are endemic to Thailand and their populations are not abundant. The research includes two species (A. mekongensis and A. versicolor) that were previously studied in terms of flower size, phenology, and floral visitors (Jirabanjongjit et al 2024), and one more, A. lycioides, which has not been previously studied.

The title of the work suggests a study with a methodological design and results that include for each of the species the recording and observation of different parameters of the flowers, which allow comparing their morphology, such as the size of petals, stamens, and anthesis and phenology. ; as well as very specific observations of floral visitors, their frequency and visiting behaviors. However, many floral characteristics, phenology, and floral visitors were previously published for A. mekongensis and A. versicolor, two of the species indicated in the study (Jirabanjongjit et al 2024). This generates a lot of confusion since these two species are presented in the method as if they were going to be studied in the present work, and then it is mentioned that there is already data on them. The writing becomes even more confusing since the data on phenology and floral visitors of these two species in the results section are presented as if they had been obtained in the present study.

I consider that the authors should make it clear in the introduction that some aspects of the reproductive biology of A. mekongensis and A. versicolor are already known and that the present study aims to know some aspects of the reproductive biology of A. lycioides, as well as know some aspects of the histochemistry of the three species, to finally compare the floral morphology, phenology and histochemical characteristics between the three species, and relate these aspects to the floral visitors.

Experimental design

The method is not sufficiently described. For example, there is no complete description of the study site, such as the type of vegetation present, environmental characteristics, when the study was carried out, etc. These are presented in the introduction, but their location should be in Materials and Methods. Likewise, the characteristics of the species included in the study are not described. A good description of A. mekongensis and A. versicolor is given in Jirabanjongjit et al (2024). Furthermore, the method for obtaining floral characteristics, such as phenology, flower size, and flower life span, is not adequately described, nor is there a good description of how the registration of floral visitors to the plant was carried out. A. lycioides.

As mentioned, the results are confusing, since data from A. mekongensis and A. versicolor are included as if they had been obtained in the present study. The presentation of quantitative data of the three species in Tables would greatly help to better understand the study. The only table presented contains the qualitative results of the histochemical analysis of the three species.

Validity of the findings

New information about the biology of any species is an important contribution to science and is more relevant if they are rare species with a restricted distribution. The manuscript is interesting since it aims to present information on some characteristics of the flowers and floral visitors of three endemic species of the genus Argyreira. However, El Ms presents new information on the reproductive biology of one species (A. lycioides), and histochemical data on the three species mentioned. That, as I mentioned before, is confusing.

Additional comments

The present form that the Ms has is not clear and robust. Various comments and suggestions are made throughout the writing

It seems to me that it is an interesting study due to the characteristics of the species included in the research, but very forced since there is no quantitative data to support the comparisons and discussion that is made between these three species.

Reviewer 3 ·

Basic reporting

The way to show about Figure 3-5 is good, but some arrows, text in the figure and scale bars are too small to read.

Experimental design

no comment

Validity of the findings

no comment

Additional comments

Abstract
The first sentence: I would like to know the location. Is it widely seen? Is it only in part of area?
...while the third (A. lycioides) has not yet been evaluated. -> What is not evaluated? The purpose of this study will gradually become clear as we read through the abstract, but I would like clarification from beginning of a sentence.
We investigate key floral characters -> We investigate floral visitors, key floral characters...

Figure 1: Genus names (Argyreia) can be abbreviated using the first initial and a period

---

## Round 0.2 · Minor Revisions

Thank you for submitting your carefully revised manuscript.
Reviewer #2 has suggested minor revisions. Additionally, I have the following minor comments for your consideration:

I don’t recall any discussion of the mating system of these three plants. Are they all self-compatible? If nothing is known, I think that should be stated (perhaps in the Methods when the species are being described).

Figure 1 – it would have been nice to have scale bars for flowers in these photos. I’m not sure if it would be possible to add them?

Figure 3 caption – check word spacing (missing spaces)

To me, the reported difference in floral longevity was most striking between A. versicolor and A. mekongensis. Although not required, I would be interested to know whether you have any ideas of what selective factors may have led to these differences; such ideas could be added to the Discussion.

Abstract
L 16 Please use past tense for methods (“investigated”)
L 26 “which contribute” to “which may contribute” (since your study did not confirm such contribution(s))

·

Basic reporting

This version of the manuscript had a notable improvement in many sections. The specific paragraphs and sentences that the authors were asked to improve, to give a better explanation, to revise or to consider any changes, as well as the incorporation of new information in tables were considered by the authors. We thank the authors for considering the comments and suggestions made in the first version. I think this version of MS is very robust. However, I detected some small details in the text indicated below, but that does not mean I can recommend its publication once the authors make these small modifications.

Minor concerns

L79. Change “:” for “;”

L179. Insert “,” . ), commonly

L178-181. Suggestion: Both A. mekongensis and A. versicolor are woody twiners (Staples & Traiperm, 2010) commonly found growing on wild plants, A. mekongensis around 1.5 m from the ground and A. versicolor around 5 m from the ground.

L313-314. Delete “r.”; 36-40 h and 15-16 h

L328-329. I preferred the conventional alphabetic sequence: “including wasps (Vespidae; Figure 3I), an unknown bee (Anthophila; Figure 3H), several ants (Formicidae), Cinnyris jugularis sunbirds (Nectariniidae; Figure 3J),”

Figure 4. Delete “r” in axis

L385 observed; rounded conical (Figure 4F6d), bell-shaped (Figure 6D), rounded cylindrical (Figure 4G6F), convex (Figure 6F), and

L396. rounded cylindrical (Figure 7B), globose (Figure 7B), obovoid (Figure 7D), and pyriform (Figure 7D)

L442. floral size? or flower size?

References. Many references have an estrange command between the title and journal name. Please review.

L667. Delete “.” After "?"

L801. Delete “.” After "?"

L855 and L858. Invert the order

L886. Change Odormediated for Odor-mediated

L894. Delete “.” After "?"

L982. Complete “Biotropica 191-205”.

L1005. Delete “.” After "?"

Experimental design

The experimental design and description of the study species was improved. which makes the study repeatable

Validity of the findings

Validity of the findings the studio makes a great contribution to knowledge of the reproduction of these three species which are endemic with populations extremely small. Besides the standard knowledge of reproductive ecology, the authors include histochemical studies of flowers, which enhance the information about these species

Additional comments

This version of the manuscript had a notable improvement in many sections. The specific paragraphs and sentences that the authors were asked to improve, to give a better explanation, to revise or to consider any changes, as well as the incorporation of new information in tables were considered by the authors. We thank the authors for considering the comments and suggestions made in the first version. I think this version of MS is very robust. However, I detected some small details in the text indicated below, but that does not mean I can recommend its publication once the authors make these small modifications.

Minor concerns

L79. Change “:” for “;”

L179. Insert “,” . ), commonly

L178-181. Suggestion: Both A. mekongensis and A. versicolor are woody twiners (Staples & Traiperm, 2010) commonly found growing on wild plants, A. mekongensis around 1.5 m from the ground and A. versicolor around 5 m from the ground.

L313-314. Delete “r.”; 36-40 h and 15-16 h

L328-329. I preferred the conventional alphabetic sequence: “including wasps (Vespidae; Figure 3I), an unknown bee (Anthophila; Figure 3H), several ants (Formicidae), Cinnyris jugularis sunbirds (Nectariniidae; Figure 3J),”

Figure 4. Delete “r” in axis

L385 observed; rounded conical (Figure 4F6d), bell-shaped (Figure 6D), rounded cylindrical (Figure 4G6F), convex (Figure 6F), and

L396. rounded cylindrical (Figure 7B), globose (Figure 7B), obovoid (Figure 7D), and pyriform (Figure 7D)

L442. floral size? or flower size?

References. Many references have an estrange command between the title and journal name. Please review.

L667. Delete “.” After "?"

L801. Delete “.” After "?"

L855 and L858. Invert the order

L886. Change Odormediated for Odor-mediated

L894. Delete “.” After "?"

L982. Complete “Biotropica 191-205”.

L1005. Delete “.” After "?"

---

## Round 0.3 · accepted · Accept

Reviewers' comments have been appropriately addressed in the revised manuscript, and I recommend this revised manuscript for publication.